# Epigenetics and Control of Tumor Angiogenesis in Melanoma: An Update with Therapeutic Implications

**DOI:** 10.3390/cancers16162843

**Published:** 2024-08-14

**Authors:** Gerardo Cazzato, Nicoletta Sgarro, Nadia Casatta, Carmelo Lupo, Giuseppe Ingravallo, Domenico Ribatti

**Affiliations:** 1Section of Molecular Pathology, Department of Precision and Regenerative Medicine and Ionian Area (DiMePRe-J), University of Bari “Aldo Moro”, 70124 Bari, Italy; n.sgarro1@studenti.uniba.it (N.S.); giuseppe.ingravallo@uniba.it (G.I.); 2Innovation Department, Diapath S.p.A., Via Savoldini n.71, 24057 Martinengo, Italy; nadia.casatta@diapath.com (N.C.); carmelo.lupo@unibg.it (C.L.); 3Engineering and Applied Science Department, University of Bergamo, 24127 Bergamo, Italy; 4Section of Human Anatomy and Histology, Department of Translational Biomedicine and Neuroscience, University of Bari Medical School, 70124 Bari, Italy; domenico.ribatti@uniba.it

**Keywords:** melanoma, epigenetics, angiogenesis, tumor progression, methylation profile, histone modifications, miRNA

## Abstract

**Simple Summary:**

Angiogenesis is a crucial process in the progression and metastasis of melanoma. Recent research has highlighted the significant role of epigenetic modifications in regulating angiogenesis. This review comprehensively examines the current understanding of how epigenetic mechanisms, including DNA methylation, histone modifications, and non-coding RNAs, influence angiogenic pathways in melanoma, with some important therapeutic approaches.

**Abstract:**

Angiogenesis, the formation of new blood vessels from pre-existing ones, is a crucial process in the progression and metastasis of melanoma. Recent research has highlighted the significant role of epigenetic modifications in regulating angiogenesis. This review comprehensively examines the current understanding of how epigenetic mechanisms, including DNA methylation, histone modifications, and non-coding RNAs, influence angiogenic pathways in melanoma. DNA methylation, a key epigenetic modification, can silence angiogenesis inhibitors such as thrombospondin-1 and TIMP3 while promoting pro-angiogenic factors like vascular endothelial growth factor (VEGF). Histone modifications, including methylation and acetylation, also play a pivotal role in regulating the expression of angiogenesis-related genes. For instance, the acetylation of histones H3 and H4 is associated with the upregulation of pro-angiogenic genes, whereas histone methylation patterns can either enhance or repress angiogenic signals, depending on the specific histone mark and context. Non-coding RNAs, particularly microRNAs (miRNAs) further modulate angiogenesis. miRNAs, such as miR-210, have been identified as key regulators, with miR-9 promoting angiogenesis by targeting E-cadherin and enhancing the expression of VEGF. This review also discusses the therapeutic potential of targeting epigenetic modifications to inhibit angiogenesis in melanoma. Epigenetic drugs, such as DNA methyltransferase inhibitors (e.g., 5-azacytidine) and histone deacetylase inhibitors (e.g., Vorinostat), have shown promise in preclinical models by reactivating angiogenesis inhibitors and downregulating pro-angiogenic factors. Moreover, the modulation of miRNAs and lncRNAs presents a novel approach for anti-angiogenic therapy.

## 1. Introduction

One of the most aggressive and deadly types of skin cancer is melanoma, which, despite improvements in early detection and treatment, is becoming increasingly prevalent worldwide, and treating advanced-stage melanoma effectively is still challenging [1,2]. While melanoma has a well-documented history of genetic mutations, including those in NRAS and BRAF genes among others [3,4], epigenetic changes have been increasingly understood to play a pivotal role in the etiology and progression of the disease [5].

The term “epigenetics” describes heritable modifications to gene expression that take place without alterations to the underlying DNA sequence [6,7]. These modifications are mediated by several different mechanisms, such as histone modifications, DNA methylation, and the action of non-coding RNAs, or “microRNAs” [8,9,10]. Since UV radiation and other environmental exposures are known risk factors for melanoma, epigenetic modifications are important because they are dynamic and subject to influence from a variety of environmental factors [8]. We have learned so much about epigenetic mechanisms since the first description by the developmental biologist, geneticist, and embryologist, C. H. Waddington, that we are now able to classify them. When DNA methylation occurs in promoter regions, for instance, it results in gene silencing because it adds a methyl group (CH3) to the cytosine residues in CpG dinucleotides [9]. Apoptosis evasion and uncontrolled cell development are favored by abnormal DNA methylation patterns found in melanoma, specifically the hypomethylation of oncogenes and the hypermethylation of tumor suppressor genes [10]. Histone modifications include many post-translational processes, which affect chromatin structure and gene expression, including methylation, acetylation, phosphorylation, and ubiquitination [11]. Histone changes in melanoma can activate or inhibit gene transcription, which influences tumor behavior. For example, the EZH2 enzyme’s methylation of histone H3 on lysine 27 (H3K27me3) has been linked to the development of melanoma and the silencing of genes [12]. Non-coding RNAs are molecules that are essential for controlling the expression of genes but do not encode proteins. The two main groups of RNAs are miRNAs and long non-coding RNAs (lncRNAs); miRNAs usually suppress the expression of genes by binding to complementary regions on target mRNAs, which causes translational inhibition or mRNA destruction [13]. Specific miRNA dysregulation in melanoma has been connected to tumorigenesis, metastasis, and treatment resistance. On the other hand, lncRNAs can influence gene expression through a variety of processes, such as transcriptional interference and chromatin remodeling, which aids in the onset and spread of melanoma [14].

Angiogenesis, which is defined as the formation of new blood vessels from the pre-existing vascular bed, is widely accepted in the literature as the fundamental mechanism underlying the development and progression of tumors [15]. The idea of the angiogenic switch is the shift in the tumor microenvironment (TME) from a latent avascular phase to a vascular phase with an unbalanced ratio of pro- to anti-angiogenic factors [16,17].

The angiogenesis process is regulated through different vascular genes and growth factors, and the impacted genes’ epigenetic states also play a role. Consequently, it is evident that epigenetic regulation, tumor angiogenesis, and progression are strictly related. In this review, we examine the most recent research on the epigenetics and control of angiogenesis in melanoma, discuss published clinical applications, and attempt to draw out potential future treatment approaches.

## 2. DNA Methylation in Melanoma

A crucial epigenetic modification known as DNA methylation occurs when a CH3 group is added to the cytosine ring 5 carbon in CpG dinucleotides, forming 5-methylcytosine [18]. A family of enzymes known as DNA methyltransferases (DNMTs), which includes DNMT1, DNMT3A, and DNMT3B, catalyzes this process [19]. DNA methylation primarily takes place in the CpG islands, which are areas with a high frequency of CpG sites that are normally found in the promoter regions of genes. The addition of methyl groups can change the way transcription factors bind to genes and generate proteins affecting the structure of chromatin [20]. Gene silencing generally results from the methylation of promoter CpG islands, whereas active gene transcription is frequently linked to unmethylated promoters [19,20].

Aberrant DNA methylation patterns are important for carcinogenesis in the setting of melanoma [21]. The hypermethylation of tumor suppressor gene promoters can lead to their deactivation, removing key growth inhibitory signals and allowing for uncontrolled cellular proliferation. Global hypomethylation, on the other hand, has been linked to genomic instability and oncogene activation [19,22]. Particularly, CDKN2A, which codes for the tumor suppressor proteins p16INK4a, p14ARF, and RASSF1A, a regulator of the cell cycle and apoptosis, is often hypermethylated in melanoma [23]. The prevalence of CpG island methylation in primitive melanoma samples, its associations with important clinicopathological features, and its prognostic significance in terms of disease-free survival (DFS) and overall survival (OS) were the main goals of a retrospective observational analysis carried out by some authors [22].

The study examined DNA methylation in a set of 170 FFPE melanoma samples using the methylation-specific multiplex ligation-dependent probe amplification (MS-MLPA), and the findings showed that at least one gene had methylation in more than half of the patients (55%). Among these one, the gene of Adenomatous Polyposis Coli (APC) (16%), Cadherin 13 (CDH13) (16%), Estrogen Receptor 1 (ESR1) (14%), Cyclin-dependent kinase inhibitor 2A (CDKN2A) (6%), and Ras association domain-containing protein 1 (RASSF1) (5%), were the most methylated, while Retinoic acid receptor (RARB) accounted for 31% of the most frequently methylated genes; these results indicated a noteworthy pattern of methylation-induced gene silencing in melanoma. Furthermore, methylation was more typically detected in older patients and those with increased Breslow thickness and it was also linked with the presence of mitosis, ulceration, rapid-growing melanomas, progressing stages of the disease, and TERT mutations, which are parameters traditionally associated with a poor prognosis. Finally, Kaplan–Meier curves were used to analyze survival data, showing that patients with methylated genes had worse outcomes with a lower likelihood of overall survival and a higher risk of disease recurrence [22].

## 3. DNA Methylation and Angiogenesis in Melanoma

Using a mouse model, Monteiro et al. examined the effects on angiogenesis, molecular pathways, and epigenetic variables of the change from nonmetastatic 4C11^−^ cells to metastatic 4C11^+^ cells caused by the loss of P53 expression. Experiments performed in the chick embryo chorioallantoic membrane (CAM) demonstrated that 4C11^−^ cells developed smaller, localized tumors, while 4C11^+^ cells formed larger, more invasive, and vascularized tumors. In 4C11^+^ cells, a pathway analysis revealed increased angiogenic signaling [fibroblast growth factor 2 (FGF2), platelet-derived growth factor (PDGF), and (VEGF), accompanied by the activation of genes such as VEGFC and angiopoietin 2 (ANGPT2) and the epigenetically hypomethylated promoters of SIX1, ANGPT2, and VEGFC, that were studied as potential biomarkers of progression [23].

Beginning with the function of thrombospondin-1 (TSP1), a well-known inhibitor of angiogenesis that is frequently silenced in a variety of cancers, including melanoma, other authors [24] investigated whether it would be possible to reverse the silencing of the TSP1 gene by treating melanoma cells with 5-Aza-deoxycytidine (5-Aza-dC), a demethylating agent, thereby reducing the angiogenesis by decreasing DNMT1 levels. In five distinct melanoma cell lines, the therapy effectively demethylated the TSP1 promoter region, restoring TSP1 expression. It is interesting to note that this demethylation only affects melanocytic cancer cells, while normal human melanocytes showed no change in TSP1 levels.

Based on the promising findings, the researchers moved to in vivo studies, employing a mouse model implanted with A375 melanoma cells. Prior to implantation, they administered 5-Aza-dC therapy to some of these cells and the results were quite intriguing because, even before there was a discernible change in tumor size, there was a considerable drop in the number of blood vessels in tumors from pretreatment cells as opposed to those from untreated cells.

Subsequent studies showed that mice treated with 5-Aza-dC had an average tumor volume that was 55% less than that of untreated controls. To highlight the significance of TSP1, the researchers used shRNA to knock down its expression, which increased tumor-induced angiogenesis by 68%. These results clearly showed that demethylation-induced TSP1 re-expression efficiently suppresses angiogenesis and tumor development. Lastly, the authors examined the methylation state of the TSP1 promoter in both treated and untreated cancers to comprehend the clinical significance of their findings. It was discovered that only 17% of the tumors from treated mice had partly methylated TSP1 promoters, compared to 67% of tumors from untreated mice. TSP1 methylation was found to be substantially more common in melanoma samples compared to non-malignant nevi, which is consistent with the idea that the methylation silencing of TSP1 is a typical characteristic of melanoma [25].

Lastly, with reference to the connection between epigenetic regulation and angiogenesis, a study [25] examined the tissue inhibitor of matrix metalloproteinases-3 (TIMP3), considering that promoter hypermethylation frequently results in decreased TIMP3 levels in melanoma. In total, 43 patients were studied, whose melanoma had spread to a lymph node, to better understand the role of TIMP3 in the disease. They evaluated TIMP3 expression and its effects on several variables, including patient survival rates, blood vessel density, and macrophage infiltration. In total, 74% of patients had significantly lower TIMP3 expression and this reduction was found to have a noteworthy impact on tumor biology. TIMP3, as an inhibitor of angiogenesis, was confirmed by the finding that there was an inverse relationship between TIMP3 expression and blood vascular density, which indicates that lower levels of TIMP3 were linked to an increase in blood vessels. Nonetheless, the lack of statistical significance (*p* = 0.369) in the association between TIMP3 expression and macrophage infiltration implies that TIMP3’s impact on macrophage activity may not be as strong or may include more intricate interactions that call for additional research. Interestingly, the TIMP3 gene promoter’s methylation status also offered important information. In fact, in 18% of the cases under analysis, the methylation of the promoter was linked to a decrease in TIMP3 expression, and it was also substantially associated with worse clinical outcomes. In comparison to patients lacking promoter methylation, those whose tumors had methylated TIMP3 promoters had worse 5-year DFS (*p* = 0.024) and OS (*p* = 0.034) estimates. This discovery emphasizes how epigenetic changes affect gene expression and the prognosis of patients. These findings showed that improving TIMP3 expression by demethylation treatments can possibly prevent tumor angiogenesis and enhance melanoma patients’ prognoses. Consequently, TIMP3 is a promising target for therapeutic intervention as well as a useful prognostic marker, opening the door to future melanoma treatments that will be more effective [25].

## 4. Histone Post-Translational Modifications in Melanoma

Histone post-translational modifications (PTMs) are critical regulatory mechanisms that play a central role in the dynamic regulation of chromatin structure and function, impacting gene expression, DNA repair, replication, and recombination [25]. Histones, the protein components around which DNA is wound to form nucleosomes, can undergo various PTMs, including methylation, acetylation, phosphorylation, ubiquitination, and ADP-ribosylation [26]. These modifications occur on the histone tails, which protrude from the nucleosome core, and on the globular domains of histones. Each modification can alter the interaction between histones and DNA or between histones and other nuclear proteins, thereby influencing chromatin compaction and the accessibility of the underlying genetic material [27].

Overall, the complex network of histone PTMs in melanoma represents a crucial aspect of the disease’s epigenetic landscape. By altering the chromatin structure and gene expression, these modifications contribute to melanoma’s aggressive behavior and resistance to conventional therapies [28]. Ongoing research aims to elucidate the precise mechanisms by which histone PTMs influence melanoma progression and to develop targeted therapies that can modulate these epigenetic marks, offering hope for more effective treatments for this challenging malignancy.

For instance, acetylation is one of the major histone changes linked to melanoma; when histone tails are acetylated, especially on lysine residues, the chromatin structure becomes more flexible, which allows transcriptional activation [28]. Histone acetyltransferases (HATs) and histone deacetylases (HDACs) have been shown to exhibit aberrant activity in melanoma [29]. Indeed, tumor growth and resistance to apoptosis can result from the silencing of genes that control cell cycle arrest and apoptosis, as well as tumor suppressor genes, when HDACs are overexpressed in melanoma. Consequently, to restore the expression of these fundamental genes, HDAC inhibitors have been investigated as therapeutic agents in the treatment of melanoma [30].

Histone methylation is another important PTM in melanoma, especially at the arginine and lysine residues [31]. In melanoma cells, the equilibrium between HMT and HDM is frequently upset, resulting in abnormal patterns of gene expression. Histone H3 trimethylation at lysine 4 (H3K4me3), for example, is typically linked to active transcription, whereas trimethylation at lysine 27 (H3K27me3) is linked to gene repression. Variations in these methylation marks in melanoma may cause tumor suppressor genes to be suppressed or oncogenes to be activated. The enzyme EZH2, which catalyzes H3K27me3, is often overexpressed in melanoma and is linked to a dismal prognosis. One possible treatment approach that is being researched is using certain inhibitors to target EZH2 [32].

Histone phosphorylation, shown by histone H2AX at serine 139 (γ-H2AX), is an essential component of the DNA damage response. The presence of γ-H2AX in melanoma is suggestive of genomic instability and DNA damage, two characteristics common to malignancy [33]. Melanoma cells’ resistance to treatment and their ability to effectively repair DNA damage through histone alterations are related. Comprehending the dynamic relationship between histone phosphorylation and the DNA damage response in melanoma may result in innovative therapeutic strategies that augment the efficacy of drugs that damage DNA [34].

Melanoma is also related to histone ubiquitination, which is the process of adding ubiquitin molecules to histones [35]. For instance, transcriptional repression is linked to the monoubiquitinating of histone H2A at lysine 119 (H2Aub1), whereas active transcription is linked to the monoubiquitinating of histone H2B at lysine 120 (H2Bub1). Genes involved in the spread and development of melanoma can have their expression altered by the dysregulation of histone ubiquitination [36].

The upregulation of HDAC6 was able to promote the expression of PDL-1, determining the activation of the PD1/PDL-1 axis, and was responsible for the depletion of T lymphocytes [37]. Furthermore, HDAC6 was also shown to interact with Tyrosine-protein phosphatase non-receptor type 1 (PTPN1), mediating the increase in protein levels independently of histone-deacetylating activity. PTPN1 was shown to increase cell proliferation and migration, with a reduction in apoptosis through the activation of the extracellular signal-regulated kinase 1/2 (ERK1/2) pathway. Finally, the HDAC6/PTPN1/ERK1/2 axis leads to an increase in matrix metallopeptidase 9 (MMP9), leading to melanoma metastasis [38]. The importance of Sirtuin 5 (SIRT5) in the progression of melanoma, including uveal melanoma, has been demonstrated [39].

HDACs impact angiogenesis by many means, most prominently by modulating the stability and functionality of hypoxia-inducible factor 1 alpha (HIF-1α), a crucial transcription factor that triggers the activation of genes related to angiogenesis in low-oxygen environments. HDACs can either increase or decrease the transcription of pro-angiogenic factors through the deacetylation of HIF-1α and other associated proteins [40,41].

## 5. MiRNAs in Melanoma

Over the past decade, a new category of cellular regulators known as non-coding RNAs has emerged [42]. These endogenous non-coding RNAs, particularly microRNAs, range in length from 19 to 22 nucleotides and their role is to modulate gene expression by silencing or degrading mRNA [42,43,44]. It is important to note that a single miRNA can inhibit multiple mRNA targets [41,44]. In fact, miRNAs are responsible for regulating more than 60% of human genes and, in the context of cancer, miRNAs can function as both tumor suppressors and oncogenes, depending on various factors such as tissue type, cellular environment, and target genes [45].

Recent research [43] has extensively documented the frequent release of exosomes by tumor cells, playing a critical role in various stages of tumor development and metastasis. The transport of different biomolecules, such as proteins, RNAs, and lipids, from tumor cells to the surrounding environment is largely responsible for these events [43].

Other clinical and experimental studies [42,44] have discovered a multitude of exosomal miRNAs in melanoma. These miRNAs have come under intense scrutiny as researchers strive to unravel their functions and molecular mechanisms in the context of melanoma. The focus, in this field, is mainly on miRNAs derived from melanoma exosomes [46]. Vignard et al. [44] conducted a study that analyzed the role of melanoma-derived exosomes in CD8þ T cells. Their results revealed that these exosomes can suppress T cell responses by inhibiting T cell receptor (TCR) signaling and reducing the secretion of cytokines and granzyme B; consequently, this result hinders the cytotoxic capabilities of the cells. Their investigation revealed that exosomes contain a high concentration of specific miRNAs, including hsa-miR-3187-3p and hsa-miR-498. The regulation of TCR signaling and tumor necrosis factor alpha (TNFα) secretion is attributed to hsa-miR-122, hsa-miR-149, and hsa-miR-181a/b. Based on these findings, it can be inferred that these microRNAs play a significant role in these processes.

Intercellular communication involves the participation of miRNAs [42,47], which play a crucial role in aiding melanoma-derived exosomes to facilitate tumor immune evasion, and, for this reason, these miRNAs could potentially serve as a valuable target for therapeutic intervention [48].

To determine the most suitable miRNA candidates or targets, it is necessary to analyze both experimental data and patient information. However, the intricate regulatory network of miRNA poses challenges for miRNA-based therapies.

Despite the obstacles that remain in the utilization of exosomal miRNAs for diagnosing melanoma and implementing effective treatments, there is ongoing encouragement and numerous endeavors to overcome these challenges.

The focus on miRNA regulation by researchers garners significant interest and sparks innovation. Additionally, the development of engineered exosomes as carriers holds promise for a secure, effective, and precise treatment option for melanoma.

## 6. MiRNAs as Regulator of Angiogenesis in Melanoma

In melanoma, specific miRNAs, such as miR-210, miR-221/222, and miR-126, have been identified as key modulators of angiogenesis. miR-210 (sometimes defined as ‘hypoxamir’), for example, is upregulated in hypoxic conditions typical of the TME, and it promotes angiogenesis by targeting regulatory pathways involved in endothelial cell migration and tube formation [49]. Furthermore, hypoxia induces the production of HIF, a typical hallmark of the TME caused by high cell proliferation exceeding the blood supply. miR-210 promotes tubular structure development and endothelial cell migration by downregulating the Ephrin-A3 (EFNA3) gene, which normally inhibits angiogenesis [48]. miR-210 influences the HIF-1α pathway, leading to the upregulation of VEGF; so, miR-210 enhances HIF-1α activity, further amplifying the VEGF-signaling cascade and boosting angiogenesis [49].

Furthermore, miR-210 influences the expression of crucial components in the pathways that generate reactive oxygen species (ROS) and mitochondrial metabolism, specifically targeting the mitochondrial respiration-related gene ISCU (iron–sulfur cluster scaffold), which modifies cellular metabolism to adapt to hypoxic circumstances and promotes angiogenic activities [50,51]. Finally, miR-210 modulates matrix metalloproteinases (MMP) expression, promoting the invasive and migratory capabilities of these cells, enhancing angiogenesis [52].

Regarding miR-221/222, this miRNA target c-Kit is a tyrosine kinase receptor involved in various cellular processes, including survival, proliferation, and the migration of endothelial cells. By downregulating c-Kit, miR-221/222 reduce the levels of downstream signaling molecules such as Akt and ERK, which are critical for endothelial cell function and angiogenesis. This suppression disrupts the normal signaling required for endothelial cell migration and tube formation, thereby influencing angiogenesis [53]. miR-221/222 also target the cell cycle inhibitors p27Kip1 and p57Kip2, stimulating endothelial cell proliferation and cell cycle progression by downregulating these inhibitors [54]. Additionally, miR-221/222 influence the PI3K/AKT signaling pathway, which is essential for angiogenesis. For example, increased AKT signaling leads to the regulation of PTEN, a negative regulator of the PI3K/AKT pathway, by miR-221/222, which in turn promote angiogenesis and endothelial cell survival [55].

Finally, miR-221/222 indirectly affect HIF-1α and VEGF pathways, promoting angiogenesis and contributing to extracellular matrix (ECM) remodeling by targeting genes involved in the degradation and reorganization of the ECM, facilitating tumor invasiveness and spreading [54,55,56,57,58,59].

Conversely, miR-126 enhances the responsiveness of endothelial cells to VEGF, promoting the expression of VEGF receptor 2 (VEGFR2), and enhancing angiogenesis [59]. miR-126 targets and downregulates SPRED1, a negative regulator of the Ras/ERK signaling pathway, which is involved in cell proliferation and migration, and with this inhibition, miR-126 leads to enhanced Ras/ERK signaling, promoting endothelial cell survival and angiogenesis [60]. Additionally, miR-126 influences the phosphoinositide 3-kinase (PI3K)/AKT pathway by targeting inhibitors of this pathway, and, in this fashion, supports endothelial cell survival, growth, and migration, all of which are critical for angiogenesis [58]. Additionally, miR-126 is involved in regulating inflammatory responses in the TME, specifically targeting and inhibiting the expression of vascular cellular adhesion molecule-1 (VCAM-1), which is involved in inflammation and leukocyte adhesion. MiR-126 promotes angiogenesis and the advancement of the tumor by lowering VCAM-1 levels, which also lessens the recruitment of inflammatory cells to the tumor site [57]. Like other miRNAs, miR-126 also exerts an influence on MMPs, further promoting angiogenesis [61].

Regarding other types of miRNAs, researchers identified a convergent and cooperative miRNA network that promotes melanoma spread, and three miRNAs were found to be endogenous promoters of angiogenesis, colonization, and metastatic invasion in melanoma: miR-1908, miR-199a-5p, and miR-199a-3p. The heat shock factor DNAJA4 and apolipoprotein E (ApoE) are the convergent targets of these miRNAs.

DNAJA4 and ApoE are essential for suppressing metastasis, and through its interaction with melanoma cell LRP1 and endothelial cell LRP8 receptors, cancer-secreted ApoE reduces invasion and metastatic endothelial recruitment (MER). DNAJA4 enhances ApoE’s anti-metastatic action by promoting ApoE expression, and it was discovered that the expression levels of these miRNAs and ApoE correlated with the outcomes of human metastatic progression, suggesting their potential use as prognostic indicators.

Metastasis inhibition was achieved therapeutically by treating the melanoma cells with locked nucleic acids (LNAs) targeting miR-1908, miR-199a-5p, and miR-199a-3p. Moreover, the administration of these LNAs therapeutically greatly inhibited the spread of melanoma. This highlights the convergent cooperativity of these miRNAs in generating metastasis, and highlights their dual functions in cancer, both cell-intrinsic and cell-extrinsic [62,63].

## 7. Therapeutic Approaches

A complex approach is employed in therapeutic intervention for the epigenetic dysregulation of the angiogenesis control in melanoma to interfere with the genetic and epigenetic pathways that promote blood vessel development. As already stated above, epigenetic modifications play a major role in the intricate interplay of signaling pathways and genetic variables that govern angiogenesis in melanoma [64]. A class of epigenetic drugs known as DNMTis targets DNA methylation processes, providing a therapeutic approach that inhibits the progression of melanoma by blocking DNMTs, which can reverse aberrant methylation and restore silenced genes. DNMTis function by incorporating into the DNA during replication and trapping the DNMT enzymes, leading to their degradation; this results in a decrease in global DNA methylation levels, especially at CpG islands in promoter regions, which can restore the expression of tumor suppressor genes. The primary DNMTis used in cancer therapy are azacitidine (5-azacytidine) and Decitabine (5-aza-2′-deoxycytidine), respectively; 5-azacytidine integrates into RNA and DNA, thereby affecting both transcription and DNA methylation, while Decitabine specifically incorporates into DNA, leading to hypomethylation and gene reactivation [65]. In melanoma, DNMTs can remove methyl groups from DNA and restore the activity of tumor suppressor genes, such as p16INK4a, RASSF1A, and MGMT, which are frequently repressed in melanoma. Reactivating these genes can impede the course of the cell cycle, stimulate programmed cell death, and bolster the systems that repair DNA. This, in turn, diminishes tumor growth and enhances the effectiveness of other treatments [66]. DNMTis may inhibit the tumor’s ability to develop new blood vessels by reactivating anti-angiogenic genes, such as TSP-1 TIMPs. Furthermore, DNMTis can boost the production of cancer/testis antigens and other antigens associated with tumors, hence enhancing the ability of melanoma cells to trigger an immune response. By enhancing the potency of immunotherapies, such as immune checkpoint inhibitors (such as anti-PD-1 and anti-CTLA-4 antibodies), it is possible to stimulate a stronger and more effective immune response against tumors [62]. Histone deacetylase inhibitors (HDACis) inhibit the process of deacetylation, resulting in an increase in acetylated histones and a chromatin state that is more relaxed and permissive to transcription; indeed, HDACis can suppress HDACs, which in turn allows for the reactivation of genes that also oppose the angiogenesis process. HDACis increase the expression of the Cyclin-dependent kinase inhibitor p21, resulting in the arrest of the cell cycle at the G1 phase. In addition, they have the ability to stimulate the production of pro-apoptotic proteins such as BAX and decrease the levels of anti-apoptotic proteins like BCL-2, thus shifting the equilibrium in favor of programmed cell death [67]. HDACis have been discovered to influence the TME and the immune response to melanoma [68]; they have the ability to improve the presentation of major histocompatibility complex (MHC) molecules and co-stimulatory molecules on the outside of cancerous cells, hence increasing their visibility and susceptibility to attack by cytotoxic T lymphocytes. In addition, HDAC inhibitors might decrease the release of immunosuppressive cytokines and chemokines, thus enhancing the effectiveness of immune checkpoint inhibitors, which have significantly transformed the treatment of melanoma in recent years [69]. Preclinical investigations and early-phase clinical trials have yielded promising results about the effectiveness of HDACis in treating melanoma; some research shows that HDACis can effectively decrease the growth of melanoma, cause tumors to shrink, and amplify the therapeutic benefits of other treatments like BRAF and MEK inhibitors [70]. Nevertheless, the advancement of HDACis in the therapeutic treatment of melanoma encounters numerous obstacles. For example, an important challenge is the identification of prognostic indicators that can aid in the selection of patients who are most likely to derive benefits from HDAC inhibitor therapy, and also the emergence of resistance to HDAC inhibitors is a significant worry, which requires the investigation of combination therapies and sequential treatment regimens to maintain their effectiveness [71].

Regarding the use of DNMTis and HDACis and the angiogenetic process, it is important to underscore that both can affect cancer angiogenesis by inhibiting pro-angiogenic genes such as VEGF and/or eNOS [72]. In melanoma, DNMTis, such as 5-azacytidine and Decitabine, incorporate into DNA and inhibit DNMT activity, leading to DNA hypermethylation that can reactivate tumor suppressor genes and anti-angiogenic factors. Among these, as previously discussed, the TSP1 gene can be re-expressed, counteracting pro-angiogenic factors in the TME [24]. Furthermore, DNMTis can reduce the expression of VEFG, as well as lower the angiogenic capacity of melanoma cells through the demethylation and reactivation of genes that inhibit VEGF production [73]. Finally, DNMTis can interfere with the HIF-alpha pathway by demethylating genes that negatively regulate the stability and activity of HIF-alpha, resulting in the reduced transcriptional activity of target genes involved in angiogenesis [74].

HDAC inhibitors can prevent histone deacetylation, ensuring that the chromatin structure remains accessible, and can be bound by transcriptional factors that promote the expression of genes with anti-angiogenic function [75]. One of the most important mechanisms consists in the acetylation of HIF-alpha by HDACs such as Vorinostat and Panobinostat, which mediate the polyubiquitination and degradation of this factor [76,77]. Following this, the transcriptional activity of HIF-alpha is reduced, mediating the reduction in levels of VEGF and other pro-angiogenic factors. HDACs also reduce the expression of MMPs, as well as downregulating angiopoietins and FGFs [78].

Table 1 summarizes the epigenetic drugs employed in the modulation of angiogenesis in melanoma.

## 8. Conclusions

Research into melanoma angiogenesis underscores the pivotal roles of genetic mutations, epigenetic modifications, and microRNA regulation in tumor progression. These findings highlight promising avenues for therapeutic intervention, particularly through targeted approaches aimed at reversing aberrant epigenetic marks and modulating angiogenic pathways. Further exploration of these mechanisms holds significant potential for enhancing treatment strategies and improving outcomes in melanoma patients.

## Figures and Tables

**Table 1 cancers-16-02843-t001:** Summary of the current drugs employed in melanoma.

Family of Drugs	Name	Mechanism of Action	Current Stage of Use
DNMTis	5-Azacytidine	Integrate into DNA and RNA restore the activity of p16INK4a, RASSF1A, and MGMT re-active anti-angiogenic genes	In vitro In vivo Early clinical trials
	Decitabine	integrate into DNA hypometilation tumor suppressor genes re-activation	In vitro In vivo Clinical trials
HDACis	Vorinostat	hyperacetylation of histones increase the expression of the CKI p21 stimulate the production of pro-apoptotic proteins (BAX) and decrease the levels of anti-apoptotic proteins like BCL-2 hyperacetylation of histones increase the expression of the CKI p21 stimulate the production of pro-apoptotic proteins (BAX) and decrease the levels of anti-apoptotic proteins like BCL-2	In vitro In vivo Clinical trials
	Panobinostat		In vitro In vivo Clinical trials

## Data Availability

New data are contained in the manuscript.

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
