# Peer review of "Epigenetics and Control of Tumor Angiogenesis in Melanoma: An Update with Therapeutic Implications"

_cancers, 2024, doi:10.3390/cancers16162843_

Round 1

Reviewer 1 Report

Comments and Suggestions for Authors

The review of Cazzato et al. summarizes the research regarding the epigenetic regulation of melanoma angiogenesis. It fits well with the topic of the Special Issue: Diagnosis and Treatment of Cutaneous Melanoma of the Cancers Journal. As angiogenesis is crucial for the progression of melanoma it is important to search for novel

therapeutic approaches that target this process. The Authors quite comprehensively summarized the recent literature in the topic of epigenetic regulation of angiogenesis. However, there are few major aspects that in my opinion should be addressed

Here are my more detailed comments:

1)    Line 111-128 – Within the 3 paragraphs there is no reference included. Even if those paragraphs describe the last cited paper it is needed to repeat the reference so that the reader knows what is the source of this data. If the Authors describe the same paper within 3 paragraphs it would be more clear to combine it into one paragraph and also shorten the text. It is too lengthy.

2)    Line 130-156 – In 3 paragraphs the Authors describe one paper that is cited only once. I recommend shortening the 3 paragraphs into one and focusing mainly on the epigenetic regulation aspect of the angiogenesis regulation.

3)    Line 167 -183 – The same comment here. The 2 new paragraphs start and there is no reference. Is it still the continuation of the paper [22]? If yes please include citations and combine the paragraphs.

4)    Line 192-212 – Lack of Reference

5)    Line 285 – What does “Through the utilization of mimics” mean? Please rephrase

6)    Line 297-303 – Those two paragraphs are redundant and should be shortened into 1-2 sentences. I also would like the Authors to explain what they mean by “Various natural anti-cancer measures”

7)    Line 355-356 – What does “(…) study including the in vivo selection of human cancer cell populations” mean? I would like to ask Authors to rephrase this sentence.

8)    Line 354-358 – Lack of reference

9)    Line 360-365 – Lack of reference

10) Line 446 – The whole Reference section needs to be revisited by the Authors. The style of references needs to be unified, for example, it looks like references 1-24 have different formatting than references 25- onwards. Also references 65-71 are probably some non-deleted formatting from the document template.

Comments on the Quality of English Language

The quality of English is mostly good, some sentences need to be rephrased as I commented above.

Author Response

Comments 1: 

The review of Cazzato et al. summarizes the research regarding the epigenetic regulation of melanoma angiogenesis. It fits well with the topic of the Special Issue: Diagnosis and Treatment of Cutaneous Melanoma of the Cancers Journal. As angiogenesis is crucial for the progression of melanoma it is important to search for novel

therapeutic approaches that target this process. The Authors quite comprehensively summarized the recent literature in the topic of epigenetic regulation of angiogenesis.

Answer n'1: Thank you very much dear Reviewer n'1 for these beatiful words; we'll do our best to improve our manuscript following your suggestions.

Reviewer n'1:1)    Line 111-128 – Within the 3 paragraphs there is no reference included. Even if those paragraphs describe the last cited paper it is needed to repeat the reference so that the reader knows what is the source of this data. If the Authors describe the same paper within 3 paragraphs it would be more clear to combine it into one paragraph and also shorten the text. It is too lengthy.

Answer n'2: Thank you very much for this comment. We have improved this paragraph.

Reviewer n'1: 2)    Line 130-156 – In 3 paragraphs the Authors describe one paper that is cited only once. I recommend shortening the 3 paragraphs into one and focusing mainly on the epigenetic regulation aspect of the angiogenesis regulation.

Answer n'2: Thank you very much dear Reviewer. We have shortened the paragraph.

Reviewer n'1:3)    Line 167 -183 – The same comment here. The 2 new paragraphs start and there is no reference. Is it still the continuation of the paper [22]? If yes please include citations and combine the paragraphs.

Answer n'3: Done. Thank you.

Reviewer n'1: 4)    Line 192-212 – Lack of Reference.

Answer n'4: Done. Thank you.

Reviewer n'1: 5)    Line 285 – What does “Through the utilization of mimics” mean? Please rephrase.

Answer n'5: Done. Thank you.

Reviewer n'1: 6)    Line 297-303 – Those two paragraphs are redundant and should be shortened into 1-2 sentences. I also would like the Authors to explain what they mean by “Various natural anti-cancer measures”.

Answer n'6: Done. Thank you.

Reviewer n'1:7)    Line 355-356 – What does “(…) study including the in vivo selection of human cancer cell populations” mean? I would like to ask Authors to rephrase this sentence.

Answer n'7: Done. Thank you.

Reviewer n'1: 

8)    Line 354-358 – Lack of reference

9)    Line 360-365 – Lack of reference.

Answer n'8: Done. Thank you.

Reviewer n'1:10) Line 446 – The whole Reference section needs to be revisited by the Authors. The style of references needs to be unified, for example, it looks like references 1-24 have different formatting than references 25- onwards. Also references 65-71 are probably some non-deleted formatting from the document template.

Answer n'8: Done. Thanks again.

Reviewer 2 Report

Comments and Suggestions for Authors

This is a well-organised literature review evaluating the mechanisms influencing angiogenesis in melanoma with a focus on epigenetic control. The manuscript is well-written and flows smoothly. Some paragraphs lack references, we advise authors to improve this aspect. Also, please check the uniformity of the acronyms. In the therapeutic approaches paragraph should be mentioned other articles in which were reported the correlation between methylation and PD-1/PDL-1 therapy. 

Author Response

Comments 1: This is a well-organised literature review evaluating the mechanisms influencing angiogenesis in melanoma with a focus on epigenetic control. The manuscript is well-written and flows smoothly. Some paragraphs lack references, we advise authors to improve this aspect. Also, please check the uniformity of the acronyms. In the therapeutic approaches paragraph should be mentioned other articles in which were reported the correlation between methylation and PD-1/PDL-1 therapy. 

Answer n'1: Thank you. We have added the references in the right places, with a entire check of the acronyms. Furthermore, we have added some articles about the methylation and PD-1/PDL-1 therapy.

Reviewer 3 Report

Comments and Suggestions for Authors

The review ‘Epigenetic control and regulation of tumor angiogenesis in melanoma: an update with therapeutic implications' by Cazzato et al., describes the role of epigenetic in angiogenesis in melanoma. 

The review is well written, however, the title of the review does not fully reflect the content. The authors switch between a couple of paragraphs on angiogenesis and a couple of paragraphs on melanoma. The link between the two is not adequate. 

Here are some major concerns:

1. The title indicates that the focus of the review is angiogenesis, however, this is missing at times. For example, in section 4, the link between histone PTMs and angiogenesis is simply missing. The authors focus instead on the pathology of melanoma, which is not the focus of the review. The same is true in section 7. The authors describe mostly therapeutic approaches for melanoma in general- specifically when they discuss HDACi. The link to angiogenesis should be the main focus of the review. Or the title needs to be revised- and the word angiogenesis removed.

2. The introduction/abstract is not adequate at times. The authors give examples of miR-9, -182, and -210 in the abstract. Yet only -210 is discussed in the review. The others not. The same is true for TSP-1 and TIMP3 (in the introduction).

3. Please provide a section on ICI and angiogenesis. There are currently only a couple of sentences on ICI and melanoma therapy in general.

4. Please stress the importance/impact of HDACs in angiogenesis. This link is not clear. 

5. Please add a Table to include the 'real' angiogeneic therapeutic approaches to support section 7.

Minor comments:

1. In line 378, DNMTis have already been introduced in line 92 and don't need reintroduction

2. The same is true for HDACis in line 405. They ahve been introduced in line 235.

3. Line 169, therapy is usually not administered to cells. The words "cells were treated" or something to the likes of that is used. 

Comments on the Quality of English Language

The title does not reflect the content of the review. There are some sentences that are not properly worded (although grammatically correct). 

Author Response

Reviewer 3 The review ‘Epigenetic control and regulation of tumor angiogenesis in melanoma: an update with therapeutic implications' by Cazzato et al., describes the role of epigenetic in angiogenesis in melanoma. The review is well written, however, the title of the review does not fully reflect the content. The authors switch between a couple of paragraphs on angiogenesis and a couple of paragraphs on melanoma. The link between the two is not adequate.

Answer Dear Reviewer n’3, first thank you very much for your words useful to improve the quality of our manuscript. We have improved the informations about the epigenetic control of ANGIOGENESIS in melanoma to enhance the link between these two topics.

Reviewer 3

  1. The title indicates that the focus of the review is angiogenesis, however, this is missing at times. For example, in section 4, the link between histone PTMs and angiogenesis is simply missing. The authors focus instead on the pathology of melanoma, which is not the focus of the review. The same is true in section 7. The authors describe mostly therapeutic approaches for melanoma in general- specifically when they discuss HDACi. The link to angiogenesis should be the main focus of the review. Or the title needs to be revised- and the word angiogenesis removed.

Answer We have added some other informations about the therapy related to epigenetics and angiogenesis in section 7 but also regarding PTMs in melanoma, although no informations were available about the PTMs in regulation of angiogenesis in melanoma. For this reason, we have decided to modify the title of our manuscript in “Epigenetics and Control of Tumor Angiogenesis in Melanoma: An Update with Therapeutic Implications”.

Reviewer The introduction/abstract is not adequate at times. The authors give examples of miR-9, -182, and -210 in the abstract. Yet only -210 is discussed in the review. The others not. The same is true for TSP-1 and TIMP3 (in the introduction).

Answer We have changed both abstract and added TSP1 and TIMP3 section in the “3. DNA methylation and angiogenesis in melanoma” section.

Reviewer Please provide a section on ICI and angiogenesis. There are currently only a couple of sentences on ICI and melanoma therapy in general. Please stress the importance/impact of HDACs in angiogenesis. This link is not clear.

Answer We have improved this link between HDACs and angiogenesis.

Reviewer: 5. Please add a Table to include the 'real' angiogeneic therapeutic approaches to support section 7.

Answer: Done.

Reviewer 1. In line 378, DNMTis have already been introduced in line 92 and don't need reintroduction 2. The same is true for HDACis in line 405. They ahve been introduced in line 235. 3. Line 169, therapy is usually not administered to cells. The words "cells were treated" or something to the likes of that is used.

Answer n’7: Done.

Round 2

Reviewer 1 Report

Comments and Suggestions for Authors

I thank the Authors for incorporating corrections to most of my comments. I still detected a few minor issues that I believe should be improved.

1. The Authors still failed to include the reference in paragraphs 127-139. Please add the citation, not only the surname if the first Author. 

2. I see that in the new version, the Authors included a new table with the drugs that targets angiogenesis. However , they fail to refer to the Table in the whole text. I would like to ask the Authotrs to refer to the Table in the proper part of the text where it fits. 

3. Please re-visit Reference 36, something went wrong with the formatting of the citation. 

A more general comment for the Authors is that in the future it would make a Reviewer's work easier and faster if all the new changes are marked in the new version of the text, not only some of them.

Author Response

Comments 1: I thank the Authors for incorporating corrections to most of my comments. I still detected a few minor issues that I believe should be improved.

Answer n'1: Thank you very much.

Comments 2: 2. I see that in the new version, the Authors included a new table with the drugs that targets angiogenesis. However , they fail to refer to the Table in the whole text. I would like to ask the Authotrs to refer to the Table in the proper part of the text where it fits. 

Answer n'2: Done. Thank you.

Comments 3: 3. Please re-visit Reference 36, something went wrong with the formatting of the citation. 

Answer n'3: Done, thank you.

Comments 4: A more general comment for the Authors is that in the future it would make a Reviewer's work easier and faster if all the new changes are marked in the new version of the text, not only some of them.

Answer n'4: Dear Reviewer, thank you and sorry for this mistake.

Reviewer 3 Report

Comments and Suggestions for Authors

I thank the authors for addequatley addressing the comments. However, one comment remains. 

The authos added a table to show the durgs used in agiogenesis treatment. However, the purpose of a table is to clarify and simplify the information. The table is not very informative in its current format. Please expand on the table. For example, please add a column to briefly describe the mechanism of action, whether it is used in vivo or in vitro or in clinical trials, the cancers it was used to treat, was it used in a combination, etc.. 

Comments on the Quality of English Language

The language is still not scientifically sound. It has improved, but it needs editing by a scientific native speaker. 

Author Response

Comments 1: The authos added a table to show the durgs used in agiogenesis treatment. However, the purpose of a table is to clarify and simplify the information. The table is not very informative in its current format. Please expand on the table. For example, please add a column to briefly describe the mechanism of action, whether it is used in vivo or in vitro or in clinical trials, the cancers it was used to treat, was it used in a combination, etc.. 

Answer n'1: Done. Thank you.